# “Me Dieron Vida”: The Effects of a Pilot Health Promotion Intervention to Reduce Cardiometabolic Risk and Improve Behavioral Health among Older Latinos with HIV

**DOI:** 10.3390/ijerph19052667

**Published:** 2022-02-25

**Authors:** Daniel E. Jimenez, Elliott R. Weinstein, John A. Batsis

**Affiliations:** 1Department of Psychiatry, University of Miami Miller School of Medicine, Miami, FL 33136, USA; 2Department of Psychology, University of Miami, Coral Gables, FL 33146, USA; erw73@miami.edu; 3Division of Geriatric Medicine, Department of Medicine, University of North Carolina School of Medicine, Chapel Hill, NC 27599, USA; john.batsis@unc.edu; 4Department of Nutrition, Gillings School of Global Public Health, University of North Carolina, Chapel Hill, NC 27599, USA

**Keywords:** older Latinos, HIV, health promotion, cardiometabolic risk

## Abstract

There are significant gaps in knowledge about the synergistic and disparate burden of health disparities associated with cardiovascular health issues, poorer mental health outcomes, and suboptimal HIV-care management on the health of older Latinos living with HIV (OLLWH). This pilot study sought to evaluate the feasibility and acceptability of an innovative application of an already established health-promotion intervention—Happy Older Latinos are Active (HOLA)—among this marginalized population. Eighteen self-identified Latino men with an undetectable HIV viral load and documented risk of cardiometabolic disease participated in this study. Although the attrition rate of 22% was higher than expected, participants attended 77% of the sessions and almost 95% of the virtual walks. Participants reported high satisfaction with the intervention, as evident by self-report quantitative (CSQ-8; *M* = 31, *SD* = 1.5) and qualitative metrics. Participants appreciated bonding with the community health worker and their peers to reduce social isolation. Results indicate that the HOLA intervention is an innovative way of delivering a health promotion intervention adapted to meet the diverse needs and circumstances of OLLWH, is feasible and acceptable, and has the potential to have positive effects on the health of OLLWH.

## 1. Introduction

People living with HIV (PLWH) are increasingly living into older adulthood at higher rates due to advances in both antiretroviral therapy (ART) and early HIV diagnostic tools. In 2018, older adults over the age of 50 accounted for over half (51%) of the PLWH in the United States (U.S.), and 17% of new HIV diagnoses nationally were in this group [1]. Although older adults living with HIV (OALWH) are living longer and healthier lives, a unique set of challenges associated with physical and mental health comorbidities associated with accelerated aging and other structural health disparities will be further exacerbated as the proportion of OALWH in the U.S. is expected to double over the next 15 years [2].

Certain marginalized groups such as racial, ethnic, and sexual minorities continue to experience significant HIV health disparities, particularly those living at the intersection of multiple system of oppression. Latinos account for almost ¼ of the U.S.’s prevalence of HIV as well as 29% of new HIV diagnoses nationally in 2019 [3]. Latino men who also identify as a sexual minority (e.g., gay, bisexual, men who have sex with men) experience a disproportionate number of new cases of HIV (almost 25%) compared to their non-Latino peers [3,4,5]. Similarly, Latinos with HIV face additional HIV-related health disparities associated with managing their HIV-care, such as inconsistent ART adherence [6] and sub-optimal viral suppression rates [2,7] compared to other racial/ethnic peers. Such disparities become significantly more observable among older Latinos living with HIV (OLLWH) facing greater rates of unsuppressed HIV viral load, poorer ART adherence, and more severe disease progression compared with their age-matched White-non-Latino counterparts [8,9]. Therefore, a greater emphasis on the intersections between HIV and aging is warranted, particularly among racial, ethnic, and sexual minority groups [10].

OLLWH face increased rates of age-related physical comorbidities due to a phenomenon called accelerated aging [11,12], in part as a result of the disproportionate rates of cardiometabolic diseases such as obesity, hypertension, and metabolic syndrome (MetS) [13,14,15]. Similarly, OLLWH tend to be more sedentary and engage with physical activity less often than their white counterparts [16]. Despite the Hispanic Paradox [17], this sedentary lifestyle in combination with certain HIV-related health outcomes such as Hepatitis-C co-infection or ART toxicity make OLLWH more likely to experience additional physical health challenges such as coronary heart disease, stroke, nonalcoholic fatty liver disease, and cardiovascular issues compared to white-OALWH [18,19].

Moreover, MetS has been identified as a risk factor for severe complications of COVID-19 [20,21]. Before the pandemic, insufficient physical activity was already described as a public health problem for OLLWH, since they are more likely to be sedentary and not as actively engaged in increasing their physical activity levels compared to non-Latino whites [16]. As a result of the current situation in which many people are confined to their homes, physical activity levels have drastically declined while dietary habits have remained unchanged or failed to offset this inactivity [22,23]. There is strong epidemiological evidence that a chronic sedentary lifestyle is detrimental for health [24,25]. Taken together, these data underscore the compounded and unequal burden of HIV, MetS, and COVID-19, which is associated with excess morbidity and mortality among OLLWH.

The threat of COVID-19 infection is especially frightening for OLLWH since age, underlying medical conditions, and race and ethnicity are major risk factors for a severe course and mortality [26,27]. While the initial focus has been on understanding the severe acute respiratory syndrome coronavirus 2 that causes COVID-19, there are also significant public health challenges arising from the resulting prevention and mitigation measures, as they are associated with increased cardiometabolic risk, depression, anxiety, isolation, loneliness, and decreased quality of life [22,28,29,30].

Depression and anxiety among older adults with HIV have been reported at five times the level in the general population and is fueled by stigma and the resultant loneliness and social isolation [31,32]. Social isolation and loneliness have been correlated with levels of morbidity and mortality comparable to more established biopsychosocial risk factors such as obesity, sedentary behavior, and hypertension [33,34,35]. Given that familism and social cohesion are strong hallmarks of Latino culture [36,37,38] and that Latinos have high levels of stigmatizing attitudes toward people with HIV/AIDS [39], OLLWH may experience the harmful effects of co-morbid social isolation and stigma more intensely than their non-Latino white counterparts. COVID-19 prevention and mitigation strategies such as physical distancing and shelter-in-place may worsen feelings of social isolation and loneliness [23]. The awareness of vulnerability as well as the involuntary and inescapable self-isolation generate anxiety and other psychiatric symptoms of distress [29]. While these measures are necessary to limit COVID-19 cases, it seems clear that actions taken to curb the spread of COVID-19 will, in fact, exacerbate depression, anxiety, and isolation-related vulnerabilities among OLLWH.

The prevalence of cardiometabolic diseases among OLLWH suggests that health-promotion interventions—defined as behavioral interventions that use counseling strategies to equip participants with the necessary knowledge and skills to modify and sustain a healthy diet, increased physical activity, and/or healthy weight—are well-aligned with their needs and may provide a tangible approach to address them. Health promotion is safe and has known benefits for those living with HIV, including improvements in waist circumference, hypertension, dyslipidemia, insulin resistance, depression, anxiety, and quality of life [40,41]. Medications may ameliorate some ART-related side effects; however, potential toxicities are associated with polypharmacy [42]. Health-promotion interventions that are safe in the context of COVID-19 are needed to manage and prevent the cardiometabolic, psychosocial, and physical functioning problems associated with HIV and ART and exacerbated by COVID-19 prevention and mitigation strategies. Despite the multiple physical and mental health benefits of health promotion, health-promotion interventions for OLLWH have not been extensively utilized or widely recognized as viable therapeutic treatment options, and the majority of OLLWH do not engage in health promotion on a regular basis [16]. Moreover, health-promotion interventions are highly adaptable and can be delivered in person, via the internet, mobile apps, or over the phone. Mobile phones hold promise as a health-promotion-intervention delivery method. Mobile phone-based health-promotion interventions have been used effectively to change physical activity and healthy eating behaviors through their ability to reach people on large scales, disseminate education material to participants, foster social support, and allow self-monitoring and feedback on behavior [43,44,45].

Studies have shown that one of the main factors mediating the adoption and maintenance of physical activity and the derivation of benefits in older adults is the creation of positive social dynamics [46,47,48]. Specifically, having friends or family who support participation in regular physical activity, through modeling or as a partner in physical activity, has been associated with long-term adherence [46,47,48]. The ability of health-promotion interventions to build social support may indeed be the key to their ability to reducing cardiometabolic risk and the psychosocial and physical functioning consequences of COVID-19 mitigation strategies in OLLWH.

Overall, little is known about the multiplicative and imbalanced burden of health disparities associated with cardiovascular health issues, poorer mental health outcomes, and suboptimal HIV-care management on the health of OLLWH. This exploratory pilot study sought to do just that by establishing that feasibility and acceptability of an innovative application of an already established health-promotion intervention entitled Happy Older Latinos are Active (HOLA) among a population of OLLWH. The objectives of this pilot study are two-fold: (1) to evaluate the feasibility and acceptability of HOLA among OLLWH (aged 50+) and (2) identify modifications needed in the design of a larger, ensuing hypothesis-testing study. Consistent with recommendations from biostatistical workgroups funded by the National Institutes of Health (NIH) [49], this pilot study was not powered to test a hypothesis. Rather, this pilot study is a requisite initial step in exploring an innovative application of the HOLA health-promotion intervention.

## 2. Methods

### 2.1. Recruitment

Eligible participants were recruited from two University of Miami consent-to-contact databases composed of PLWH. This database offered the authors access to participant demographics, contact information, and data associated with HIV-related risk factors, allowing for targeted recruitment based on the study’s inclusion criteria. Participants who were considered potentially eligible from the databases were contacted via phone and invited to enroll in the study should they continue to fit the study’s inclusion and exclusion criteria. The database yielded 89 participants who were potentially eligible. Of those 89, 19 were not interested in participating, with lack of time being the reason most often cited; 9 were unable to be reached; and 20 did not meet our eligibility criteria.

### 2.2. Participants

Potential subjects for this pilot study who self-identified as male and Latino, were living with an undetectable HIV viral load (viral load < 200 copies/mL), and had a documented risk of cardiometabolic disease were invited to participate in this study. Documented risk of potential cardiometabolic disease was defined as having at least two of the following criteria: (a) waist circumference ≥ 94 cm; (b) HDL-C < 18.5 mg/dL; (c) triglycerides ≥ 30.6 mg/dL or specific treatment for hyperlipidemia; SBP ≥ 130 mmHg or DBP ≥ 85 mmHg, or treatment of previously diagnosed hypertension; HbA1c < 6.5% (48 mmol/mol). Older Latinos who met our inclusion criteria were provided with an initial verbal summary of the study, and informed consent was gathered from those who were interested in participating prior to enrollment. Payments were compensated up to $75 for their participation in the study. Final sample size was 18.

### 2.3. Outcomes

The primary outcomes were feasibility and acceptability as measured by participant evaluation of the project. We also considered participation and attrition rate as an indicator of acceptability based on the notion that if the intervention was not found to be useful, the participants would discontinue their participation in the project. This approach to analysis of feasibility data follows a similar approach that was utilized in the first HOLA trial and comparable pilot trials [50,51]. Other factors assessed at baseline were: cardiometabolic risk, using waist circumference, fasting levels of HDL-C, triglycerides, systolic and diastolic blood pressure, and HbA1c; depression-symptom severity, using the nine-item Patient Health Questionnaire (PHQ-9) [52]; anxiety-symptom severity, using the Generalized Anxiety Disorder scale (GAD-7) [53]; perceived stress using the 14-item Perceived Stress Scale (PSS) [54]; perceived social support using the 12-item Multidimensional Scale of Perceived Social Support (MSPSS) [55]; and level of day-to-day discrimination faced due to their HIV status using the HIV-Stigma Scale [56].

### 2.4. Measures

#### 2.4.1. Demographics

Participant demographics were collected using an established demographics questionnaire developed by the Center for Latino Health Research Opportunities at the University of Miami. The questionnaire was adapted to include several other health-focused questions including asking participants to complete a medication list.

#### 2.4.2. Client Satisfaction Questionnaire (CSQ-8)

The Client Satisfaction Questionnaire [57] is an 8-item measure of client satisfaction with services. Responses to each item are scored from 1 (poor) to 4 (excellent). The CSQ-8 has no subscales and yields a single score measuring a single dimension of overall satisfaction. An overall score is calculated by summing the respondent’s rating (item rating) score for each scale item. Scores therefore range from 8 to 32, with higher values indicating higher satisfaction. In addition, there are two open-ended questions that allow participants to state in their own words what they liked most about the intervention and what they liked the least.

### 2.5. Intervention Procedures

Happy Older Latino Adults (HOLA) is a multi-component health-promotion intervention for Latinos in middle and older adulthood [50]. The intervention employs components from both Social Learning Theory [58] and Behavioral Activation [59] to help older Latinos bolster their physical activity and schedule in pleasant events to their day-to-day routine to stave off recurrent depression and anxiety symptoms. The entire intervention is administered by a community health worker (CHW), which allows for deeper connections between participants and the interventionist and in turn, allows the interventionist to successfully model, encourage, and promote consistent health-behavior change [60,61].

The HOLA intervention has four core components. The first component centers on two structured social and physical activation sessions where participants meet individually with the CHW for an individualized 30-min physical and social activation session in anticipation for the group exercise phase of the intervention. Next, participants engage in a group walk, three times a week for 45 min each, over the course of 16 weeks to facilitate both physical activity and social interaction between participants. The group walks consist of 6 participants and take place at a public park that is centrally located to where the group participants reside. During the cooldown phase after each walk, the interventionist helps participants identify a pleasant event that they intend to do with another person prior to the next walk session in a way consistent with behavioral activation. The final component of HOLA centers around a maintenance phase via “booster” group walks twice a month for three months post-intervention to reinforce behavior change and maintain intervention effects.

### 2.6. COVID-19 Adaptations

In March of 2020, all in-person clinical research activities were suspended due to the COVID-19 pandemic. The protocol was amended to conduct all study-related activities virtually (i.e., over the telephone). For the social and physical activation sessions, the CHW would call the individual participants and go over the material that would have ordinarily been shared in person. For the virtual walks, participants would conference call with the CHW and the other members of their group at the time of their regularly scheduled walk, and they would walk around their respective neighborhoods with the CHW leading them. At the end of each walk, the CHW would engage with the participant in pleasant-event scheduling. The “booster” group walks were also conducted over the phone. Therefore, all participants were able to receive all four components of the intervention. Post-intervention and 3-months post-intervention follow-up assessments were conducted over the telephone as well. We were unable to collect follow-up data on cardiometabolic risk factors since that required in-person blood draws. When clinical research activities were suspended, Group 1 had completed the 2 social and physical activation sessions and 14 weeks of group walks in person. The last 2 weeks of the intervention (6 group walks) along with the booster sessions (6 additional group walks) were completed virtually. Group 2 had completed 1 social and physical activation session and 5 weeks of group walks in person. The last 11 weeks of the intervention (33 group walks), the second social and physical activation session, and the booster sessions (6 additional group walks) were completed virtually. Group 3 completed all of the intervention activities virtually.

### 2.7. Clinical Trial Registration

The study was registered on Clinicaltrials.gov #NCT03839212. Registered on 8 February 2019. First participant enrolled on 1 October 2019.

## 3. Results

### 3.1. Sociodemographic Characteristics

Table 1 shows baseline clinical and sociodemographic characteristics of the sample. The average age of the participants was 60.5 years (*SD* = 6.2). A majority (66.7%) had a high-school education or below. The sample was diverse with respect to country of origin—Cuba, Colombia, Honduras, U.S. (including Puerto Rico), and Nicaragua—with participants living in the U.S. for an average of 30.6 years (*SD* = 11.0). All preferred to speak Spanish. The majority of participants (55.5%) identified as a sexual minority (gay or bisexual); seven identified as straight or heterosexual; and one did not specify his sexual orientation. Participants reported moderate levels of depression-symptom severity (*M* = 14.3, *SD* = 4.2), perceived stress (*M* = 34.9, *SD* = 7.1), and HIV-related stigma (*M* = 105.2, *SD* = 18.0).

### 3.2. Feasibility and Acceptability

Fourteen participants (77.8%) completed the post-intervention and 3-months post-intervention follow-up assessments. That group did not have a chance to do either the social and physical activation sessions with the CHW or the in-person group walks before all clinical research activities were suspended due to the pandemic. Participants attended 77% of the sessions. This includes the in-person walks as well as the virtual walks. The attendance rate of the virtual walks was 94.6%.

The average score of the CSQ-8 [57] was 31 (*SD* = 1.5). Participants answered the two short-answer questions on the CSQ-8 [57]. Answers to the short-form questions reinforced what the participants had endorsed in the prior quantitative questions. Their responses were transcribed. From their project evaluations, two themes emerged. First, many of the participants were socially isolated, and the group walks offered a way to connect to others who had similar lived experiences. One participant stated, “Me dieron vida”, which translates to, “You gave me life”, when talking about the effect that the intervention had during a time of quarantine and social distancing. Another participant explained that what he liked most about the program was “meeting other people like me…walking with my group, they were all very friendly.” An additional participant went a step further saying, “We have all become friends and know that we can count on each other.”

Second was the connection that the CHWs were able to build with the participants. One participant stated, “What I liked most about the program was the way the health promoter reached the group”, while another stated, “Orieta (the CHW) united us all”, highlighting the ability of CHW to connect with participants and facilitate trusting bonds between group members. A summary of each theme and the participants’ qualitative feedback is presented in Table 2.

## 4. Discussion

Our intervention results indicate that an innovative health-promotion intervention—such as HOLA—can be adapted to meet the diverse needs and circumstances of OLLWH; is feasible and acceptable; and has the potential to have positive effects on the health of OLLWH. Currently, health-promotion interventions for OLLWH have not been extensively utilized or widely recognized as viable therapeutic treatment options, and the majority of OLLWH do not engage in health promotion on a regular basis [16]. This pilot study is the first step in developing a health-promotion intervention that can be implemented on a broad scale basis.

The results from the program evaluation were extremely positive. Cultural values and beliefs were included in the formative and development phases of the intervention to ensure that it addressed the different socio-cultural influences that have a role in the health of Latinos. We emphasized positive cultural attributes and motivated health promotion by highlighting Latino cultural values, such as the importance of community (communidad), respect (respeto), and trust (confianza). Findings indicated that the participants who received the intervention reported noticeable benefits, both quantitatively and qualitatively, from their participation in the program. For example, those who received the intervention reported that they benefited in terms of social isolation, loneliness, and stigma. Prior studies have shown that one of the main factors mediating the adoption and maintenance of physical activity and the derivation of benefits in older adults is the creation of positive social dynamics [62,63]. Specifically, having friends or family who support participation in regular physical activity, through modeling or as a partner in physical activity, has been associated with long-term adherence [46,48]. The ability of health-promotion interventions to build social support may indeed be the key to their ability to affect health outcomes.

The use of CHWs was an effective and culturally acceptable means of reaching the population with health information and motivating health behaviors. Since CHWs possess an intimate understanding of the cultural norms and values inherent to a specific community, they are particularly primed for success in promoting health and health outcomes with cultural competency. In the present context, the CHW is seen as an authority figure. If a participant agrees to meet with the CHW, then the participant will keep the appointment out of respect for the CHW’s time and authority. Moreover, older Latinos living with HIV may feel more comfortable divulging personal information to someone from their own community, allowing them to communicate with study personnel in a manner with which they feel comfortable (e.g., preferred language). This person-centered approach fosters trust and overall enthusiasm for the project.

### 4.1. Lessons Learned

Based on the quantitative and qualitative results from the project evaluation, we have gained greater insight into the value of our intervention in supporting OLLWH. First, social distancing does not mean spiritual distancing. Confusion and disruption to daily life were paramount at the beginning of the COVID-19 pandemic. Most clinical research activities were suspended across the nation, and we were unsure as to when they would resume. During this time of great uncertainty, study personnel let participants know that we were with them in spirit. Staff shared their fear, anxiety, stress, frustration, and confusion and learned that maintaining a personal connection to the participants, albeit virtually, was of utmost importance. It is essential that researchers continue to think innovatively about how to bridge the digital divide and embrace low-technology options such as conference calls in order to stay connected to their participants—no matter what it takes.

Second, we learned that 10 participants (55.5%) had flip phones, which forced us to confront the digital divide. The digital divide has been well-documented. Older Latinos are less likely to have access to or use high-quality internet compared to their non-Latino white or Black counterparts [64,65,66]. Due to social distancing guidelines and COVID-19 related regulations/restrictions, the internet has become a lifeline, allowing many to stay socially connected [67]. This increasing reliance on the internet can create or exacerbate known health disparities, which are particularly important considering the high prevalence and incidence of social isolation and loneliness and their impact on functioning, and physical and mental health, in this population.

Third, we learned that the virtual tools serve to complement—not replace—the in-person services. The attrition rate of 22.2% was higher than the 15% that we had originally predicted based on other pilot studies [50,51]. The four participants who were lost to follow-up were from the third group, which had formed but had not had a chance to meet in-person before all clinical research activities were suspended due to the pandemic and the COVID-19 adaptations had been implemented. Unfortunately, the last group did not have a chance to form a cohesive bond like the prior groups had. We believe that if the group had received an opportunity to meet in-person and establish a connection, then we would have not seen as much attrition.

Fourth, we learned that COVID-19 is exacerbating social isolation and loneliness among OLLWH. The threat of COVID-19 infection is especially frightening for OLLWH since age, underlying medical conditions, and race and ethnicity are major risk factors for a severe course and mortality [26,27,68]. Prevention and mitigation strategies (i.e., physical distancing, shelter-in-place, etc.) help to attenuate adverse outcomes from COVID-19; however, these approaches may have unintended consequences on the physical and mental health of OLLWH, given that this group tends to have restricted social networks due to HIV-related stigma and ageism [68].

### 4.2. Limitations

Several limitations of this study are worth noting. First, our small sample size limited our ability to test a hypothesis regarding intervention effects. Nonetheless, our results have proven to be promising, and provide evidence that an adequately powered, randomized control trial is warranted. Second, the two short-answer questions asked what participants liked and disliked about the intervention, thus potentially limiting nuanced feedback from the participants. Third, the sample was all Latino men and, hence, we are unable to understand whether similar findings would be obtained among Latina women living with HIV. Fourth, we did not measure reasons for attrition among participants who were lost to follow-up. Finally, COVID-19 was a historical threat to the internal validity of the study. The suspension of all in-person clinical research due to the pandemic forced us to adapt the intervention and conduct the study virtually. While this adaptation allowed us to stay connected with our participants at a time of great uncertainty and fear, this meant that not everyone received the intervention delivered in the same way.

## 5. Conclusions

To our knowledge, this study is the first of its kind to assess the acceptability and feasibility of an adapted health-promotion intervention to improve cardiometabolic and mental health among a population of OLLWH. Relatively low attrition rates, high attendance of weekly sessions, and significant participant satisfaction as indicated by both quantitative and qualitative measures demonstrated the overall positive response of this intervention among a small group of OLLWH. OLLWH who received the intervention reported that they benefited in terms of social isolation, loneliness, and HIV-related stigma, which, we anticipate, improved overall mental health among this group during a particularly psychologically taxing time—the COVID-19 pandemic. Overall, HOLA is potentially successful in improving the mental health of OLLWH because of the authors’ intentional incorporation of cultural values and beliefs in the development of the intervention (e.g., utilization of the CHW model). Based on the success of this pilot study, future research should continue to explore how a culturally informed health intervention to improve physical health and reduce social isolation can be further adapted to support other marginalized older Latinos.

## Figures and Tables

**Table 1 ijerph-19-02667-t001:** Baseline clinical and sociodemographic characteristics.

Sociodemographic Characteristics	Sample (N = 18)
Age, years	60.5 (*SD* = 6.2)
Relationship Status	
% Single/never married	61.1 (n = 11)
% In a domestic partnership	5.6 (n = 1)
% Married	11.1 (n = 2)
% Separated	5.6 (n = 1)
% Divorced	16.7 (n = 3)
Education	
% No formal education	5.6 (n = 1)
% 1st to 8th grade	33.3 (n = 6)
% Some high school	5.6 (n = 1)
% High school/General Educational Development (GED)	22.2 (n = 4)
% Some college	22.2 (n = 4)
% College degree	11.1 (n = 2)
Employment	
% Employed	22.2 (n = 4)
% Unemployed	77.8 (n = 14)
Race	
% White	72.2 (n = 13)
% Black	11.1 (n = 2)
% Other	16.7 (n = 3)
Country of Origin	
% Cuba	38.8 (n = 7)
% Colombia	22.2 (n = 4)
% Honduras	16.7 (n = 3)
% U.S. (including Puerto Rico)	16.7 (n = 3)
% Nicaragua	5.6 (n = 1)
Years in the U.S.	30.6 (*SD* = 11)
Preferred Language	
% Spanish	100 (N = 18)
Sexual Orientation	
% Straight or heterosexual	38.8 (n = 7)
% Gay	33.3 (n = 6)
% Bisexual	22.2 (n = 4)
% Did not specify	5.6 (n = 1)
Cardiometabolic Risk Factors	
Waist Circumference (cm)	101.3 (*SD* = 14.8)
HDL-C (mmol/L)	46.8 (*SD* = 17.1)
Triglycerides (mmol/L)	176 (*SD* = 77.1)
Systolic Blood Pressure (mmHg)	130.6 (*SD* = 13.1)
Diastolic Blood Pressure (mmHg)	76.8 (*SD* = 10.3)
HbA1c (%)	5.4 (*SD* = 0.48)
Psychosocial Functioning	
PHQ-9	14.3 (*SD* = 4.2)
GAD-7	4.2 (*SD* = 3.5)
Perceived Stress	34.9 (*SD* = 7.1)
Social Support	4.7 (*SD* = 0.99)
HIV-related Stigma	105.2 (*SD* = 18.0)

**Table 2 ijerph-19-02667-t002:** Participant Feedback.

Themes	Illustrative Quotations
Overcoming social isolation, loneliness, and stigma via the group walks	“Me dieron vida.”
	“I can interact with other people; this is hard for me because I am shy.”
	“We have all become friends and know that we can count on each other.”
	“What I liked most about the program was meeting other people like me … walking with my group, they were all very friendly.”
	“What I liked most about the program was the group walks and being comfortable enough to talk to everyone.”
Connection with the community health worker (CHW)	“Orieta (the CHW) really listens to what I have to say.”
	“Orieta (the CHW) united us all.”
	“What I liked most about the program the way the health promoter reached the group.”

## Data Availability

Data and associated documentation from Center for Latino Health Research Opportunities (CLaRO) funded research projects will be made available to users under a data-sharing agreement that provides for: (1) a commitment to using data to advance the objectives of CLaRO and not to identify any individual participant; (2) a commitment to securing the data using appropriate computer technology; and (3) a commitment to destroying or returning the data after analyses are completed.

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
