# Peer review of "“Me Dieron Vida”: The Effects of a Pilot Health Promotion Intervention to Reduce Cardiometabolic Risk and Improve Behavioral Health among Older Latinos with HIV"

_ijerph, 2022, doi:10.3390/ijerph19052667_

Round 1

Reviewer 1 Report

Introduction: Aging, Healthcare disparity, Social isolation are mentioned in generic terms and without context of current pandemic. We are two years into pandemic. All three are dramatically impacted by a pandemic that requires social isolation and likely triggers (not researched; but you have awareness of it in your study design) greater health concerns for sexual and gender minority population. It would indeed be relevant to the readers, to gain better insight with a "current relevant introduction". There are many components of aging: increased disease symptom complexity and frailty. We also talk a lot about polypharmacy and multimorbidity in this context. Perhaps a reinvigorated introduction to this vital article will help the readers.

Introduction: What is the connection between social isolation and cardiovascular health? Is there evidence that "virtual walks" or "virtual cardiovascular activities aka peloton" can have a measured benefit? If yes, perhaps, some positive inferences can be taken for the SGM population.

Introduction: The premise for the health of older latinos seems to be rather unfocused. Concrete examples would be helpful for the reader. 

Rephrase this sentence. Don't use the phrase "responsible". It implies blame.

Intro pg 1 Line 30-31: In 2018, older adults over the age of 50 were responsible for 17% of new HIV diagnoses within the United States (U.S.) and account for just over half of the prevalence burden for HIV cases nationally (CDC, 2020). Define the sexual minority.

What is implied by etc? If you are referring to the entire spectrum of sexual minority population (LGBTQIA+), please clarify (later in the results you talk of gay and bisexual study participants only and during recruitment patients self-identified as male). Etc is not appropriate. I understand this topic is vital and under-researched, but please include additional citations. 

 Intro pg 1 Line 41-42 : Latino men  who also identify as a sexual minority (e.g., gay, bisexual, etc.) experience almost 25% of 42 new HIV diagnoses despite making up less than 10% of the entire adult population (6). 

Recruitment/Participants: Did the recruitment questionnaire ask for a) legal sex and b) sex assigned at birth? If all participants are grouped together, clinical disease outcomes are likely to be inappropriately inferred. 

Methods: pg 3, Line 106-108: Potential subjects for this pilot study who self-identified as male and Latino, were living with an undetectable HIV viral load (viral load <200 copies/mL) and had a documented risk of cardiometabolic disease were invited to participate in this study. 

Relationship Query/Table 1: That is not aligned with the SGM population being studied, especially the hispanic SGM! For that reason, %Single/never married is 61% (n=11). Within this group, participants might be sexually active and not interested in marriage/divorce life model. That needs to be taken into consideration for study design and clinical outcomes. The more pertinent questions would be about sexual well-being, social health and how it impacts cardiovascular outcomes.

No other comments. Study design and data interpretation excellent. All the best for advancing your project. 

Author Response

On behalf of my co-authors, I would like to thank you for the constructive and interested comments and the opportunity to resubmit our manuscript. We are pleased to read that you felt that our study design and data interpretation were “excellent.” We have revised our paper significantly to address your comments and questions. Attached, we provide a guide to our revisions. Revisions in the text are in red.

Thanks to your comments, we believe that we have produced a very strong paper that over time will become highly cited. We would welcome the opportunity for further development of the manuscript as you see fit.

Reviewer 2 Report

This is a pilot study sought to evaluate the feasibility and acceptability of an innovative application of a health promotion intervention to reduce cardiometabolic risk and improve behavioural health among older Latino’s with HIV. This study has several gaps, concerning several aspects:

  • There is no data on the number of participants who were considered potentially eligible, and the reasons why they were not interested in participating prior to enrolment
  • How many core components of the HOLA intervention were fulfilled before suspension of face-to-face activities, in March 2020?
  • There is no data of the initiation of the HOLA intervention
  • It’s not clear how many of the four core components of the HOLA intervention were suspended. Taking in account that the aim of the study was health promotion intervention to reduce cardiometabolic risk, and the clinical research activities were suspended together with the impossibility to collect follow-up data on cardiometabolic risk factor, how the feasibility and acceptability of the HOLA was evaluated?
  • Depriving social and physical sessions, replacing the group walks by virtual walks, and post intervention and 3-months post intervention follow-up assessments conducted over the telephone, how this adaptation of the protocol can achieve results indicating that the HOLA intervention is an innovative way of delivering a health promotion intervention, feasible and acceptable and has the potential to have positive effects on the health of OLLWH?
  • In summary, it’s not easy to interpret the findings indicating “that the participants who received the intervention reported noticeable benefits both quantitatively and qualitatively from their participation in the program”. In terms of reducing cardiometabolic risk, how the intervention (with COVID-19 adaptations) benefits quantitatively and qualitatively the participants

Minor corrections:

48th line – correct to “…among older Latino’s living with HIV (OLLWH)…”

82nd line – correct to “…on the health of OLLWH…”

90th line – correct to “…National Institute of Health (NIH)…”

183rd line – the rate of 66.6% do not matched with the rate presented in Table 1

285-291st lines – there is no data on how many groups were put together, and which was the criteria to group the participants in the program

Author Response

On behalf of my co-authors, I would like to thank you for the constructive and interested comments and the opportunity to resubmit our manuscript. We have revised our paper significantly to address your comments and questions. Attached, we provide a guide to our revisions. Revisions in the text are in red.

Thanks to you, we believe that we have produced a very strong paper that over time will become highly cited. We would welcome the opportunity for further development of the manuscript as you see fit.

Round 2

Reviewer 2 Report

The manuscript was improved. There is no more comments. Congratulations.